# Development and Applications of Embedded Passives and Interconnects Employing Nanomaterials

**DOI:** 10.3390/nano12193284

**Published:** 2022-09-21

**Authors:** Shanggui Deng, Sharad Bhatnagar, Shan He, Nabeel Ahmad, Abdul Rahaman, Jingrong Gao, Jagriti Narang, Ibrahim Khalifa, Anindya Nag

**Affiliations:** 1Department of Food Science and Pharmaceutics, Zhejiang Ocean University, Zhoushan 316022, China; 2Faculty of Life and Environmental Sciences, University of Tsukuba, 1-1-1, Tennodai, Tsukuba 305-8572, Ibaraki, Japan; 3Institute for NanoScale Science and Technology, College of Science and Engineering, Flinders University, Bedford Park 5042, Australia; 4Department of Biotechnology, School of Allied Sciences, Dev Bhoomi Uttarakhand University, Naugaon 248007, India; 5School of Food Science and Engineering, South China University of Technology, Guangzhou 510641, China; 6Department of Biotechnology, School of Chemical and Life Sciences, New Delhi 110062, India; 7Food Technology Department, Faculty of Agriculture, Benha University, Moshtohor 13736, Egypt; 8Faculty of Electrical and Computer Engineering, Technische Universität Dresden, 01062 Dresden, Germany; 9Centre for Tactile Internet with Human-in-the-Loop (CeTI), Technische Universität Dresden, 01069 Dresden, Germany

**Keywords:** embedded components, resistors, capacitors, inductors, interconnects, nanomaterials

## Abstract

The advent of nanotechnology has initiated a profound revolution in almost all spheres of technology. The electronics industry is concerned with the ongoing miniaturization of devices and as such requires packaging technologies that will make the devices more compact and resilient. 3D packaging, system in package, and system on chip are the various packaging techniques that utilize nanoscale components for their implementation. The active components of the ICs have kept pace with Moore’s law, but the passive components have proven an impediment in the race for miniaturization. Moreover, the toxic effects and nano-scale problems associated with conventional soldering techniques have entailed the active involvement of nanotechnology in the search for answers. Recent advances in these fields and the diverse nanomaterials which are being employed to resolve these issues have been discussed in detail.

## 1. Introduction

The era of nanotechnology is slowly but surely making its way across the threshold of conventional technologies. The creation and manipulation of a structure at a range of 1 nm to 100 nm refer to nanotechnology [1]. The field of electronics has already entered the nano zone since the size of integrated circuits reached below 100 nm [2]. With the 22 nm complementary metal-oxide semiconductor (CMOS) in development, nanotechnology has established itself well in the field of electronics [3]. However, apart from integrated circuits (ICs), the bulk of the components (about 80%) are discrete passive components (capacitors, resistors, inductors) and these components occupy a majority of the space on the surface of a printed circuit board (PCB) [4]. The consistently increasing demand for miniaturization and high performance at a low cost for microelectronics has considerably shifted the attention towards packaging technologies: system on chip (SOC) and system in package (SiP) being the premier among them [5]. 3D system packaging is the only way to further reduce the volume and bulk of the microelectronics products available today and this is where nanotechnology comes into the picture. The aim of nano-packaging is to develop nanoscale passives, interfaces, and interconnections, which can be compressed into highly miniaturized systems. The standard definition of nano-packaging, according to IEEE can be considered as follows: “Nano-packaging can be defined as the process of interconnecting, powering, cooling, and protecting the nanocomponents made of nanomaterials to form electronic and bioelectronic systems for greatly improved functionality and cost [6].”

Over the last sixty years, there has been exceptional improvement in the speed and reliability of the microprocessor. According to Moore’s law, the number of transistors in semiconductor devices or ICs would double approximately every two years [7]. This has more or less held true for the active components of ICs, but the passive components have been unable to follow the miniaturization trend completely because of issues with material development and fabrication technologies. To overcome the size and space demanded by the discrete passive components, an alternative technology called embedded passives is being researched for the past decades. This technique involves embedding the passives directly at the substrate level and thereby reducing the need for bulky discrete components. These passives can be used in conjunction with the latest packaging technologies to further scale down the size of electronic equipment. Even though the technology has been in the making for quite a few years, it has not yet been made for commercial applications largely due to materials and process issues [7]. Therefore, it is quite necessary to develop some materials that will permit suitable fabrication processes and will also perform well enough on electrical and mechanical criteria.

## 2. Embedded Capacitors

The capacitor is a component used to store electrical energy in an electric field. It usually stores energy in the form of electrostatic fields in between the plates. The removal of discrete capacitors from the surface and the embedding of them directly into the substrate board can lead to a reduction in size and weight as well as an increase in reliability, performance, and reduced cost. Requirements for the development of materials to fabricate embedded capacitors include a high dielectric constant (*k*), high breakdown, low dielectric loss, low leakage current, and good stability [8].

### High k Nanomaterials for Embedded Capacitors

Early attempts at the fabrication of high *k* dielectric materials yielded results in the form of ferroelectric ceramic materials, ferroelectric ceramic–polymer composites, conductive filler–polymer composites, and all organic polymer composites. Each of these had a disadvantage associated with them, each distinct in every case. In the case of ferroelectric ceramic materials, the high sintering temperature for fabrication was an issue. Examples of this category include Barium titanate (BaTiO_3_), lead zirconatetitanate (PbZrTiO_3_), etc. [9,10]. Ferroelectric ceramic–polymer composites suffered from some fabrication issues like poor dispersion of filler within the matrix and a lack of adhesion between other components in PCB. Notable examples of such composites include poly(vinylidenefluoride-co-trifluoroethylene) (P(VDF-TrFE)), magnesium niobate–lead titanate (PMN-PT) + (BaTiO_3_) [11,12]. Conductive filler—polymer composites such as Ni/PVDF composite, and LNO/PVDF (Li-Doped NiO/polyvinylidene fluoride) were reported to have very high *k* values but they also had the drawback of high dielectric loss and conductivity. Even then, these materials have a distinct advantage over ceramic/polymer composites. Organic polymer composites have been fabricated by using an organic filler material having a high dielectric constant with a polymer matrix also exhibiting the same. Examples include CuPc (copper–phthalocyanine) dispersed in P(VDF-TrFE) (poly (vinylidenefluoride-co-trifluoroethylene) [13]. These composites showed a high dielectric constant but suffered from high dielectric loss too. These composites are being further researched for embedded capacitors application.

The fabrication of high *k* dielectric constant materials has made considerable progress owing to advances in nanotechnology. Due to the unique physical and chemical properties of materials at the nanoscale, these materials may provide the solution to the issues plaguing the current technology and will point toward the future direction of electronic packaging.

To understand the dielectric properties of the polymers, a multi-core model has been proposed. According to their theory, the interface of a spherical inorganic filler particle embedded in a polymer environment consists of three discrete layers: a bonded layer, a bound layer, and a loose layer, with an electric double layer overlapping the three layers [14]. The bonded layer is the one that comes in direct contact with the filler surface, the bound layer is the one present in the interfacial region, and the electric double layer is the one where the prevailing atmosphere is similar to that of the bulk polymer. The second layer unsettles the motion of the dipoles from polar groups, consequently reducing the permittivity. The electric double layer is indirectly responsible for the reduction in dielectric constant too. Therefore, the bound layer and electric double layer properties must be controlled if the dielectric properties are to be modified. Attaching the specific organic groups to the surface of the particle can lead to the modification of the properties and interfacial area. The choice of the surface organic assembly and its various properties such as size, polarity, polarizability, and mobility could have a foremost bearing on the dielectric properties of the polymer nanocomposites [15].

Nanoparticles dispersed polymer is an excellent alternative due to its low-temperature processing and nano-sized particles allowing the formation of thinner dielectrics that are able to achieve high capacitance density [16]. The affordable processing and mechanical properties of polymers, combined with the distinct magnetic and electrical properties of nanoparticles, make polymer-nanoparticles composite a great idea. There are two types of fillers that have been researched extensively for the fabrication of nanocomposite materials: ferroelectric ceramic fillers and conductive fillers [17].

## 3. Ferroelectric Ceramic–Polymer Composites

The dielectric constant of these polymers is dependent upon the concentration of the filler used or the filler loading volume. As the filler loading volume increases, so does the dielectric constant. However, a very large loading volume also results in uneven dispersion and a lack of adhesion. The ambient processing conditions for the ceramic and unique properties of nanoparticles can lead to the fabrication of high-k materials. Barium titanate (BaTiO_3_) has been the most researched material for this category along with epoxy composites due to its high dielectric value. *k* values more than 150 to about 200 have been generated by using this ceramic-polymer composite. Cho et al. fabricated a BaTiO_3_-epoxy thin film with a thickness of 7 µm exhibiting a dielectric value of 100, 10 nF/cm^2^ capacitance, with low leakage current values of 10^−8^ A/cm^2^ [10]. A modified hydrothermal reaction method was used by Suibin et al. to synthesize BaTiO_3_ powder and the nanocomposite had a dielectric value of 19.4 at 10 kHz, and a loss factor of 0.02, with a 50% loading volume. The high dielectric value and low loss were attributed to the tetragonal crystal phase and uniform dispersion [18]. Das et al. developed a BaTiO_3_–epoxy flexible nanocomposite having a thickness from 2 microns to 25 microns exhibiting high capacitance (10–100 nF/inch^2^) and low loss (0.02–0.04) at 1 MHz. The capacitance of these films could be increased up to 500 nF/inch^2^ by the modification of composites with nanomaterials such as lead zirconatetitanate (PZT), lead lanthanum (PLZT), Zinc oxides (ZnO), Lead magnesium niobate (PMN) and PMN-PT (Lead magnesium niobate-lead titanate) [19,20]. Thermally treated BaTiO_3_ was used by Hanemann to increase the composite permittivity by inducing the phase change into tetragonal one and crystal lattice relaxation. Composites with a solid load of around 78% with a bimodal particle size distribution showed *k* values around 50 and a loss factor around 5% [21]. 

Xu et al. have optimized the BaTiO_3_–epoxy nanocomposite based on loading capacity and dielectric constant. The loading capacity above 50% has several physical and mechanical disadvantages in real applications, so the loading capacity was maintained at 50%, whereas the dielectric value was maintained at around 50. The optimized rubberized nanocomposite showed a high dielectric constant above 50, a high breakdown voltage of 89 MV/m, and low leakage current values of 1.9 × 10^−11^ A/cm^2^ [22]. Perovskite (Ca_2−x_Sr_x_Nb_3_O_10_) nanosheets have been fabricated by Osada and Sasaki using the Langmuir–Blodgett method. These nanofilms exhibited a high dielectric of greater than 200 and quite a low leakage current of 10^−7^ A/cm^2^ in films having a thickness of around 5 nm [23]. 

### 3.1. Conductive/Metallic Nanoparticles-Polymer Composites

The high loading volume of ceramic fillers in the polymer composite not only leads to an increase in *k* values but also drastically reduces adhesion and mechanical capabilities, making it less adaptable for usage on the circuit board. Another alternative is to fabricate conductive filler systems that exhibit a drastic increase in dielectric constant in the percolation systems as it nears the percolation threshold [24]. The *k* value of such composites can be given by:(1)$$\frac{k}{k_{m}}={\left[f_{c}-f\right]}^{-s}$$
where, *k_m_* = dielectric constant of the matrix,

*f* = volume fraction of the filler,

*f_c_* = volume fraction at percolation threshold,

and *s* = exponent of about 1.

The filler of metallic nanoparticles brings about unique properties in the composite. These materials have been marked as good candidates for embedded capacitor applications because of their high *k*, although the dielectric loss in these materials is very difficult to control because the conductive/metallic particles can easily form a conductive pathway in the composite as the percolation threshold is achieved. Since these composites usually require a lesser loading capacity volume as compared to polymer composites, they provide balanced mechanical properties with better adhesion. Different metallic/conductive fillers that are being used for these composites are silver (Ag), aluminum (Al), carbon black, etc. [25,26,27]. The major bottleneck for these composites is the narrow processing window between achieving a high *k* and a low dielectric loss. A wide variety of approaches have been used to improve the quality and overcome the shortcomings of these nano-composites.

Initial approaches to developing conductive filler–polymer composites utilized an epoxy–silver-based composite by using silver as a conductive filler. The dielectric constant was reported to be around 1000 and was achieved around the percolation threshold with low dielectric loss (<0.02) and good adhesion [28,29]. Lu et al. synthesized novel carbon black–polymer composites containing in situ formed silver nanoparticles. The high dielectric constant was attributed to the filing of charges at the extended interface of the interfacial polarization-based composites. The decrease in dielectric loss was credited to the Coulomb blockade effect of the Ag nanoparticles exhibiting the quantum effect of metal nanoparticles. The major effect on the properties was due to the size, size distribution, and loading level of the metal nanoparticles [30]. The different categories of conductive filler nanocomposites are discussed below. The differentiation has been performed based on the methodology that has been employed by various researchers for improving the dielectric value and reducing the dielectric loss tangent value.

### 3.2. Three Phase Nanocomposites

Three-phase percolative silver–BaTiO_3_–epoxy nanocomposite was synthesized by Qi et al. The incorporation of silver into epoxy resin led to a significant increase in the dielectric properties of the resin and BaTiO_3_ was further mixed to create a high dielectric constant polymer composite. These composites demonstrated a high dielectric constant of approximately 450 which was 110 times higher than that of the epoxy matrix, with a dielectric strength of 5 kV/mm at room temperature [31]. George and Sebastian synthesized a similar Ca[(Li_1/3_Nb_2/3_)_0.8_Ti_0.2_] O_3−*δ*_(CLNT)–epoxy–silver, three-phase composite by a two-step mixing and thermosetting technique. The addition of a 0.28 volume fraction of silver increased the relative permittivity of the composite from 8 to 142 at 1 MHz [32]. 

#### 3.2.1. Use of High *k* Polymer Matrix

The *k* values of the composite can also be increased by using a high *k* polymer matrix for the conductive filler. The high *k* polymer matrix is further reinforced by a high *k* polymer conductive filler. Lu et al. utilized this approach to generate uniformly dispersed Ag particles of around 10 nm into the polymer matrix using photochemical reduction. The number of nanoparticles in the polymer matrix was estimated at around 10% by weight. Self-passivated Al particles were then incorporated into the Ag–epoxy matrix to further improve the dielectric constant. The presence of an oxide layer on Al particles leads to the lower loss of dielectric. The composite showed a 50% increase in dielectric values as compared to Al-filled epoxy composites and the dielectric loss tangent was below 0.1 [33]. Recently, a similar approach was used by Li et al. to fabricate a Copper (Cu)–Epoxy matrix using a thermal reduction method which generated Cu nanoparticles around 100 nm with uniform dispersion. BaTiO_3_ ceramic particles with a high dielectric constant were incorporated into this Cu–epoxy matrix which enhanced the dielectric constant while maintaining the low dielectric loss compared with the BaTiO_3_/epoxy composite. Nanocomposites obtained by ex-situ techniques exhibited a comparable dielectric constant with much lower dielectric loss compared to in-situ BT/Cu–epoxy composites. The improved dielectric performance of nanocomposites was attributed to the excellent dispersion of Cu nanoparticles as well as the strong interfacial interaction between Cu nanoparticles and epoxy matrix in the in-situ Cu–epoxy matrices [34]. 

#### 3.2.2. Use of Surface Modification

Surface modification of metallic fillers via the introduction of surfactant coating can lead to a reduction in the dielectric loss of the nanocomposites. Qi et al. reported an epoxy-based composite containing silver nanoparticles having an average size of 40 nm coated with a layer of mercaptosuccinic acid (MSA) to promote the formation of Ag–epoxy while retaining the flexibility of the matrix. The dielectric constant and loss increased with the increasing filler concentration up to 22%. At the Ag concentration of 22% by volume, the dielectric constant was found to be 308 and the dielectric electric loss was quite low at 0.05 when measured at 1 kHz. The decrease in the dielectric constant after 22% was attributed to the porosity, which may have been caused by the adsorbed surfactant layer which leaves space between the Ag particles and the creation of voids that are not occupied by the polymer. Since no rapid increase in the dielectric loss was evident, it was concluded that the formation of the conducting filler network was prevented by the surfactant coating layer [28]. The dependence of dielectric value and dielectric loss on silver volume and frequency is demonstrated in Figure 1.

Ren et al. have synthesized polymer-coated silver nanoparticles by the reduction of Tollen’s reagent by means of mercaptosuccinic acid/polyethylene glycol (MSA/PEG) copolymers as reducing agents and stabilizers concurrently. The average size of Ag particles was between 10 to 120 nm, which could be controlled by changing the MSA/PEG ratio. The surface-coated Ag particles were then incorporated into the epoxy matrix leading to the formation of Ag–epoxy nanocomposites. The nanocomposites with a 25% filler volume loading exhibited a dielectric constant of 237 while the dielectric loss was below 0.08 at 1 kHz. The dielectric properties were found to be dependent upon the volume fraction of silver nanoparticles in epoxy resin [35] as shown in Figure 2. A novel idea for surface modification has been researched by Luo et al. In this method, the surface modification of BaTiO_3_ nanoparticles was performed using Ag nanoparticles, effectively making it a metal–ceramic filler. Nano Ag particles of 20 nm dimension were grown over 100 nm BaTiO_3_, which effectively prevented the Ag nanoparticle contact within the PVDF membrane, inhibiting the formation of a conducting path. The filler loading of 43.4% by volume, the *k* value of the composite was 94.3 and a dielectric loss of 0.06 was exhibited by the composite. On increasing the filler volume further, the *k* value increased to 160 at the same frequency and the loss tangent remained low at 0.11 [36].

#### 3.2.3. Use of Core-Shell Structured Fillers

The direct contact of conductive metal fillers leads to the formation of a conductive path that will direct to a higher dielectric loss at or above the percolation threshold. Core-shell structured fillers were projected as an alternative to this problem as the nonconductive shells can act as a barrier between the conductive metal fillers, thereby reducing the dielectric loss and increasing the *k* values. Xu et al. developed a high *k* polymer based on self-passivating Al as filler. The size of the Al nanoparticles was found to be around 100 nm, with an oxide thickness of about 2.8 nm as shown in Figure 3a. The layer of aluminum oxide acted as an electric barrier and significantly affected the dielectric properties of the Al-epoxy composite. A dielectric constant of 109 and dielectric loss of 0.02 at 10 kHz was observed at an 80% loading of Al by weight [37] as in Figure 3b.

Copper nanowires have been used in conjunction with PVDF polymer to formulate a nanocomposite with a high dielectric constant. The dielectric properties of these nanocomposites were compared to that of a multiwalled carbon nanotube (MWCNT) filler with PVDF polymer. It was surmised that as compared to the MWCNT-filled PVDF, this novel Cu nanowire–PVDF polymer exhibited better dielectric permittivity values with lower loss at room temperature in the frequency range of 10^−1^ to 10^−6^ Hz. The higher conductivity was attributed to the Cu core providing a high number of mobile carriers impacting the interfacial polarization whereas the lower loss was attributed to the formation of an oxide layer over the nanowires leading to suppression of the formation of the conductive path. Because of the presence of the oxide layer over the wire, this process can be classified among the core-shell structured fillers [38].

Shen et al. fabricated a polymer composite using Ag cores coated with organic dielectric shells as fillers. This dielectric shell acts as an interparticle barrier and prevents the Ag particles from coming into direct contact. It also encourages the uniform dispersion of the filler in the polymer matrix leading to *k* values of more than 300 and a low dielectric loss of 0.05 [39]. 

Recent examples of this category include gold nanoparticles of 15 nm homogenously coated with a 10 nm layer of SnO_2_ having high capacitance by Oldfield et al., carbonaceous shell coating on Ag cores by Shen et al., synthesis of TiO_2_ (Titanium oxide) nanoparticles coated with a paraffin layer by Balasubramanian et al., gold nanoparticles coated with polystyrene; and gold, silver, and titanium coated with silica shell core by Badi et al., amongst others [40,41,42,43].

Other examples of core-shell hybrid fillers other than metallic fillers are discussed below. Wang et al. have synthesized core-shell structured BaTiO_3_–polystyrene nanoparticles (BT–PS) with varying PS shell thickness and studied the influence of shell thickness on the dielectric properties. Two types of BT–PS were synthesized: with the shell thickness of 3 nm and 12 nm by controlling the polymerization time. Composites comprising core-shell nanoparticles exhibited higher dielectric constant, higher breakdown strength, and lower breakdown voltage as compared to the PS matrix. Composites filled with BT–PS with a 3 nm shell exhibited better dielectric properties as compared to BT–PS with a 12 nm shell. A maximum energy density of 4.24 J/cm^3^ obtained in BT–PS/PS films [44] was demonstrated.

Guo et al. have reported the fabrication of Novel core/shell structured multi-walled carbon nanotube/amorphous carbon (MWCNT@AC) nanohybrids which were used as fillers to improve the dielectric properties of poly (vinylidene fluoride) (PVDF)-based composites. The MWCNTs served as the core and amorphous carbon served as the shell. The composites exhibited a high possible *k* value of 5910 and dielectric loss of around 2, which is considerably better than that of MWCNT/PVDF composites [45].

Core-shell structured hyperbranched aromatic polyamide grafted barium titanate (BT-HBP) hybrid was used as a filler for a Poly (vinylidene fluoride-trifluoroethylene-chlorofluoroethylene) (PVDF-TrFE-CFE) matrix by Xie et al. to obtain a high dielectric constant value of 1485.5 at 1 kHz at a 40% loading volume. Enhanced interfacial polarization between the BT–HBP and polymer matrix was the reason for the observed high dielectric constant [46]. 

### 3.3. Embedded Resistors

The second group of passive components that are routinely studied to shrink the size of printed circuit boards and printed wiring boards (PCB/PWB) is the resistor [47]. Embedded resistors will also be able to increase the reliability and electrical performance of the circuit [48]. The integration of resistors also reduces the area requirement on the PCB/PWB, thereby potentially increasing the device functionality by placing more active components which provide gain to the system. In embedded systems, resistors are generally sheets of a material that are sized appropriately to achieve a certain resistance and whose ends are used for interconnection to other components. The resistive materials in embedded applications should have high electrical resistivity, low-temperature coefficient of resistance (TCR), and should be easy to process [48,49,50]. Apart from this, nanoparticle-based embedded resistors have another set of factors that limit their use in mainstream electronics. Yield, reliability, and a range of resistance are needed to be researched for nanoparticle-based resistors. The resistance of the material depends upon the dimension and the property of the materials. 

Another factor that is important in the fabrication of the resistors is the temperature coefficient of resistance (TCR). The rate of change of resistance with temperature is termed the TCR of the material. It is measured in units of ppm/°C and can be determined using the resistance change from some reference temperature and the change in temperature [51]. A positive value of TCR means that the resistance increases with the temperature, for example, in pure metals, whereas negative TCR means that the resistance is decreasing with the increase in temperature, such as in the case of carbon, silicon, etc. For some metal alloys, the TCR values are close to zero, meaning that the resistance does not vary with the temperature, a property that can be utilized for the formation of a high-precision resistor [52].

Traditionally, metal pastes consisting of metal particles and organic resin have been used for the fabrication of resistors. This paste is applied on the surface of the substrate and after the application of a high temperature; metal particles melt and fuse to form a film [53]. The natural properties of nanoparticles can be harnessed to fabricate low-cost resistors at low temperatures. The nanoparticle-based resistors can be classified into the following major categories: cermets, metal alloys, and carbon-filled polymers. Cermets are a mixture of ceramics and metals which can be fabricated by a number of methods such as evaporation, sputtering, co-evaporation, co-sputtering, plasma polymerization, and the mixing of metal ions in polymers [54]. Some examples of cermet resistors are discussed below.

### 3.4. Cermets

Lim et al. fabricated a SiO_2_-Pt nano-composite ceramic metal by the co-sputtering method. Cermet was fabricated by uniformly dispersing Pt particles into the SiO_2_ matrix. Resistivity values of 880 to 193,820 µΩ·cm were obtained at 3–20 mTorr, and the temperature coefficient of resistance was in the range of 383.189 to −3229.14 ppm/K [55].

Park et al. demonstrated the fabrication of Ta_3_N_5_–Ag nanocomposite thin films with near-zero temperature coefficients of resistance (TCR) fabricated by a reactive co-sputtering method which can be used as thin-film embedded resistors. The TCR value of the film was found to be near zero due to the balancing of the positive TCR value of Ag and negative TCR value of Ta–N at a resistivity higher than 0.005 Ω·cm. The co-sputtering was performed at the nitrogen partial pressure of 55% and the fabricated thin film had a resistivity of 0.0059 Ω·cm and power density of 0.94 W/cm^2^. The TCR value was found to be +34 ppm/K [56] as shown in Figure 4. Thin cermet films of thickness from 2 to 40 nm were fabricated by sputter deposition from CrSi_2_–Cr–SiC targets by a dual cathode dc S-gun magnetron by Felmetsger. The atomic ratio of Si to Cr was found to be about 2:1. At a film thickness below 2.5 nm, the temperature coefficient of resistance (TCR) was significantly increased. Cermet films with thicknesses in the range of 2.5–4 nm had sheet resistances ranging from 1800 to1200 Ω/□ and TCR values from −50 ppm/°C to near zero, respectively [57].

Liu et al. fabricated Al–Zr_2_(WO_4_) (PO_4_)_2_ (Al-ZWP) nano-cermets with the aim of controlling the coefficient of thermal expansion (CTE). SEM imaging and elemental mapping showed that the cermets consisted of uniform nanoparticles with sizes around 100 nm and Al and ZWP were found to be homogeneously dispersed in the cermets. The CTEs of the cermets were found in the range from −2.74 × 10^−6^ K^−1^ to ∼25.68 × 10^−6^ K^−1^. Depending on Al:ZWP mass ratio, the cermet could act as a capacitor or a resistor [58].

Nash et al. fabricated a series of amorphous chromium oxide (CrOx) films by dc sputtering and the sheet resistance of CrOx could be modified by increasing the level of oxygen doping. By varying the level of oxygen doping, the room temperature sheet resistance could be controlled from 28 Ω/□ to 32.6 k Ω/□. The film thicknesses were found to be in the range of 179 nm for growth in pure argon to 246 nm for growth with an oxygen partial pressure of 0.7 mTorr. Among the two contacts studied, the gold layer was found to be more favorable as compared to the silicon-niobium contact, the specific contact resistivity of chromium oxide to gold interfaces being 0.14 mΩ·cm^2^. These chromium oxide films can be used as high-Value resistors nanoscale circuits [59] in Figure 5.

### 3.5. Metal Alloys

Other classes of nanoparticle-based resistors can be termed metal alloys which can be fabricated by the deposition of thin metal films by either sputtering or electroless plating. These materials are also termed resistive alloys. Among these alloys, the most prominently used alloys are NiCr, NiCrAlSi, CrSi, TiN_x_O_y_, and TaN_x_. These resistors have also been made commercially available by some companies such as Gould and Ohmega. Nichrome (NiCr) resistors have been widely studied in terms of power handling, thermal performance, adhesion, and etching resolution. NiCr can be alloyed with Al and Si to improve thermal stability and reduce TCR, hence forming NiCrAlSi. A thin film of NiCr and NiCrAlSi can be deposited on the copper films to make an embedded resistor. Sheet resistance values of 25 to 250 Ω/sq can be obtained by varying the sheet thickness [60].

Andziulis et al. have fabricated nano-multilayer resistive films consisting of Cr–Ni–Si material. This multilayer structure consisted of 3–8 nm resistive layers with 1–2 nm sliced barrier insulator layers spaced in between to prevent the vertical coalescence of metal grains. Magnetron sputtering was used to deposit the thin films, and plasma oxidation was used to create the barrier insulator from segregated silicon. The composition of the alloy was 54% Cr, 06% Ni and 40% Si by weight so that enough silicon is present for the formation of the barrier as well as the matrix. The sheet resistance was found to be 300–550 ohm/sq. and the TCR value was in the range from ±2 ppm/K to −60 ppm/K [61]. 

TaN_x_ has been widely used as a resistive alloy for embedded resistor applications. The formation of this alloy typically consists of the reactive sputtering of Ta in a nitrogen atmosphere. These alloys can achieve stable resistivities equipped with 250 μΩ·cm with TCR of around −75 ppm/°C [62]. Sputtered TiN_x_O_y_ has been shown to offer comparatively higher resistivity up to 5 kΩ/sq with TCR of ±100 ppm/°C [63].

### 3.6. Carbon-Based Composites

Carbon-filled polymers are the latest entrants into resistor technology. The advent of nanotechnology has made these materials a possibility. Carbon nanotubes, nano-fibers, carbon black, etc. can be used as fillers in composite materials to manufacture composite resistors. For example, multiwall carbon nanotubes made by chemical vapor deposition were dispersed in an epoxy polymer by 0.01 wt.% to achieve a resistivity in the range of 10^6^–10^9^ Ω·cm range [64]. Similarly, carbon nanofibres mixed (1% by weight) with epoxy resin which were aligned using the AC field have been reported to show resistivity in the range of 10^6^–10^7^ [65]. The use of carbon black as a filler (0.12 wt.%) epoxy resin and the application of a static electricity field by metal electrodes resulted in the growth of dendrites from the anode. The resistance values of the composite ranged from 2 × 10^3^ to 10^11^ Ω·cm and could be controlled by varying the applied voltage and curing temperature [66].

### 3.7. Embedded Inductors

An inductor is a passive component present in a circuit that is used to store energy in the form of a magnetic field. Inductors form an integral part of the circuitry used in various spheres such as wireless communication (for radio frequency applications), computers, automotives, peripherals, etc. (for power supply). As such, they cover a plethora of functions from acting as amplifiers, filters, regulators to converters, and oscillators, etc. Embedded inductors require a lot of research and development on the performance front before they can be properly incorporated into packaging systems. The major hindrances in the development of embedded inductors include their low-quality factor (Q-factor), high losses, and limitations of proper miniature fabrication techniques [67,68]. 

Where these losses are of significant concern, air core inductors are used. To achieve high inductance though, magnetic cores exhibiting high permeability are required. Other important factors include high resistance to reduce eddy losses, operation at high frequencies, and low coercivity. Improvement in these factors can result in high inductance, which will in turn lead to a decrease in the number of windings, which could probably lead to miniaturization of the system. The change in properties at the nanoscale could lead to high resistance and high permeability and as such are desirable candidates as a material for fabrication.

The design of the inductor also affects the inductance capacity. Electrical, mechanical properties, and reliability issues are dependent upon the design of the inductor. There is a number of designs available for inductors that have been widely researched. The most important and thoroughly studied of these are spiral inductors because of their simple designs and high efficiency. Solenoid inductors are important for discrete components because of their high inductance and high-quality factor but they suffer from the drawbacks of inefficient packaging and high leakage. Toroidal inductors are used where low leakage currents, high inductance, and low electromagnetic interference are required. Some novel designs have been studied to overcome the traditional drawbacks of inductors; these include inductors with micro slits and surface planarization [69].

Apart from the material issues, the problems plaguing the use of nanomaterials in embedded inductors are the fabrication techniques. The growth of thin films can be achieved using techniques such as sputtering, plasma enhanced deposition, electroplating, chemical vapor deposition, etc. The patterning techniques, though, such as photolithography, micromachining, and ion milling need refining. The frequency characteristics, DC resistance, and inductance can be controlled by patterning. Some recent techniques have been developed for thin film deposition such as spin coating, spin sprayed thin films, etc. [70,71].

New classes of nanomaterials with high permeability are being researched for the fabrication of thin-film-embedded inductors. The reduction in the size might lead to a reduction in losses and stray capacitance. The materials which can operate at high frequencies with lower losses, hence increasing the inductance are the need of the hour. The various classes of magnetic nanomaterials for inductor cores are discussed below.

## 4. Alloy-Based Nanomaterials for Inductor Cores

### 4.1. Iron-Based Alloys

Iron-based alloys have been used in low-frequency and high-power inductors owing to lower losses. Nickel–iron-based alloys termed perm alloys can achieve a high range of permeability. Perm alloy is a nickel–iron magnetic alloy, containing 80% nickel and 20% iron. Molybdenum, copper, etc. are added to perm alloy to increase the permeability and reduce coercive losses. Perm alloy has low magneto crystalline anisotropy and magnetostriction which helps it to achieve high permeability. Yang et al. have developed on-chip planar inductors using nickel–ion permalloy (Ni_80_Fe_17_Mo_3_) of sub-100 nm in diameter to increase the inductance while maintaining the magnetic performance for high-frequency circuitry [72]. Zhao et al. have developed permalloy–SiO_2_ granular films with induced anisotropy using multilayer alternate sputtering with different power. The films were synthesized with different metallic volume fractions and exhibited excellent soft magnetic properties with high resistivity. Soft magnetic properties were directly dependent upon the metallic volume fraction whereas resistivity was inversely dependent upon it [72].

### 4.2. Cobalt-Based Cores

Magnetostriction can be reduced to a much higher extent by using cobalt-based alloys. Soft magnetic Co-based granular films possess properties such as high saturation magnetization, high anisotropy, and high resistivity. Soft magnetic properties of the sputtered films are only found in limited composition ranges and require low sputter pressure. Co–Al–O and Cr–Zr–O are the only oxide alloys that are suitable for inductor applications. Co–Al–O films exhibited a resistivity of 500–1000 μΩ·cm, a magnetic flux of about 80 Oe, and B_s_ of about 10 kG. With the addition of palladium, Co–Al–Pd–O, the soft magnetic properties and *H*_k_ value of the film were significantly improved, with *H*_k_ more than 180 Oe [73]. Nanogranular Co–Al–O films with a maximum resistivity of 110 mΩ·cm were fabricated by Amiri et al. These films were deposited by using pulsed dc reactive sputtering of a Co_72_Al_28_ target in an oxygen/argon ambient. The average grain size was found to be 80 nm. The effect of deposition power on resistivity and permeability of films showed that resistivity increase is associated with a decrease in coercivity but is only achievable at lower permeability and higher relaxation losses [74].

Co–Zr–O alloys exhibit preferable properties on water-cooled substrates which makes them suitable for embedded inductor applications. Ohnuma et al. reported the fabrication of Co–Zr–O nanogranular films using rf reactive sputtering in argon and oxygen ambiance. The least amount of coercivity was obtained at 55% and 70% of cobalt. The films near Co_60_Zr_10_O_30_ were reported to have a value of anisotropy at 150 Oe, saturation magnetization (*H*_k_) values of 9 kG, coercivities lesser than 3 Oe, and resistivity more than 1000 μΩ·cm. These films exhibited an excellent response of permeability to high frequency. The ferromagnetic resonance frequency values were exceeded 3 GHz [75].

Lu et al. have developed a 30 MHz power inductor utilizing nanogranular material soft magnetic material Co–Zr–O by sputtering on a polyimide substrate. The inductor was able to achieve an inductance of 470 nH and Q of 67, which was professed to double using process improvements [76]. Multilayer Co–Zr–O/ZrO_2_ thin films have been used by Yao et al. to improve the performance of the magnetic core and reduce the eddy current loss. V groove inductors were fabricated on Si substrate using sputtering techniques. The inductors exhibited an inductance of 3.4 nH in a range of 10 to 100 MHz, a dc resistance of 3.83 mΩ, a quality factor of up to at least 50, and can be applied in the manufacturing of high-power-density high-efficiency dc-dc converter devices [77].

Nitrides of Co alloys offer much wider compositional ranges for anisotropy but usually exhibit higher coercivities and lower permeabilities. Co–Al–N films were prepared by the rf-sputtering method to obtain soft magnetic properties at high frequencies by Kijima et al. The films were composed of Co nanogranules of 3–5 nm dispersed in the AlN matrix. Ferromagnetic properties were obtained in the range of compositions with cobalt content from 47–80 at.%. The films exhibited high coercivity in the range of 20–50 Oe and showed perpendicular magnetic anisotropy. The permeability of Co_80_Al_14_N_6_ film was constant up to 1 GHz and ferromagnetic resonance frequency (FMR) was at around 1.2 GHz [78].

Cao et al. fabricated Co–HfN nanogranular films with varying Co content exhibiting anisotropy over a wide composition range. The films were composed of crystallized Co nanogranules dispersed in an amorphous HfN matrix. The coercivity was in the range of 15–75 Oe. Co_58_Hf_14_N_28_ films displayed a permeability of 40 and a ferromagnetic resonance frequency (FMR) of 2 GHz [79].

### 4.3. Iron-Cobalt-Based Cores

In accordance with the Pauling—Slater curve, Fe–Co alloys have the highest magnetization compared to all the other iron alloys. These alloys exhibit a high magnetization saturation value of more than 20 kG. CoFeN films with high saturation magnetization of 24 kG, anisotropy of about 20 Oe, and low coercivity of about 1 Oe have been prepared by Sun et al. [80]. CoFeN native oxide films prepared using rf reactive magnetic sputtering in argon and nitrogen ambiances were demonstrated by Ha et.al. These films exhibited low coercivity below 1 Oe and magnetization of almost 22 kG. The anisotropy field was found to be as high as 67 Oe and consequently, the FMR value increased to 3 GHz [81]. This decrease in coercitivity was due to the addition of nitrogen. Similarly, the addition of Si or Ni can also lead to a decrease in coercivity. (CoFe)–Si–O/Si–O multilayer nanogranular magnetic thin films were produced on SiO_2_/Si substrates using inductively coupled rf sputtering by Ikeda et al. The ratio of the optimized multilayer structure in terms of the CoFe monolayer to the Si-O layer was found to be 6:1 nm. This film exhibited excellent soft magnetism performance in high-frequency regions. The permeability was found to be 200, the resistivity was found to be 2.2 mΩ·cm and FMR was about 2.86 GHz [82]. Ge at al. fabricated (Fe_65_Co_35_)_x_(SiO_2_)_1−x_ granular films by magnetron sputtering. The films were composed of Fe_65_Co_35_ particles uniformly dispersed in an amorphous SiO_2_ matrix. When the range of the *x* value was from 0.7 to 0.5, the films exhibited a small coercivity value of less than 4 Oe and resistivity up to 5.48 × 10^3^ µΩ·cm. The minimum coercivity value was found to be 1.7 Oe with a resistivity of 2.86 × 10^3^ µΩ·cm at x equaling 0.57. The real permeability of the sample was found to be 170 and the FMR frequency was about 2.4 GHz [83]. 

Kim et al. have studied the effect of the addition of boron on the magnetic properties and high-frequency characteristics Fe-Co alloys. Boron was added to Fe_7_Co_3_-based thin films and it was found that the addition of boron resulted in an increase in anisotropy, resistivity, and a decrease in coercivity without any loss of saturation magnetization. This increase was found to be dependent upon grain size, grain morphology, and orientation. Fe_55_Co_28_B_17_ displayed a ferromagnetic resonance frequency of 3.3 GHz and a linear effective permeability of 350 up to 2 GHz [84]. 

Fe–Co–Zr–O alloys have been found to possess high magnetic fluxes and as such can be used for inductor cores. Fe–Co–Zr–O films have been designed by Ohnuma et al. using reactive sputtering under oxygen–argon ambiance, utilizing Fe-Co-Zr alloy. The films consisted of base-centered cubic Fe-Co nanoparticles with nanoparticles of ZrO dispersed in between. These films exhibit a high magnetic flux density of 23 kG and low coercivity which was attributed to ZrO nanoparticles mediated inhibition of growth of Fe–Co. The process parameters of the fabrication process must be controlled strictly in order to maintain the high magnetic saturation since the increase in the value of oxides could prove unfavorable [85]. 

(Co–Fe)–Al–O nanogranular thin films have been found to have a large resistivity, a high saturation magnetization, and excellent high-frequency properties making them an attractive option for inductor cores. Sohn et al. reported Co-Fe-Al-O nanogranular thin film fabrication using rf magnetron sputtering in an argon-oxygen atmosphere to produce base-centered cubic Co-Fe particles lesser than 5 nm in the Al-O matrix. These films consistently produced high resistivity up to 400 μΩ·cm, a saturation magnetization of 16 kG, and an anisotropy field of 45 Oe. The high-frequency magnetic properties were found to be excellent which can be gauged by a high FMR value of 2.3 GHz and a real permeability value of 315 [86]. Ha et al. have studied the effect of thickness in the range of 50 to 1200 nm upon the high-frequency characteristics of (Co–Fe)–Al–O soft magnetic thin films. The coercivity and anisotropy fields of these films are dependent upon the film composition and the roughness of the surface. The resistivity of these films increased with the decrease in the grain size and increase in the oxide volume. The films were composed of a multi-phase structure consisting of α-CoFe (1 1 0), and α-CoFe (2 1 1) with minor phase CoFe–oxides and Al_2_O_3_ as the matrix. The effective permeability of this film was more than 1000 and was found unvarying up to 700 MHz while the film thickness remained under 1000 nm. The film exhibited excellent high-frequency properties owing to high resistivity and high anisotropy [87]. 

### 4.4. Ferrite Nanomaterials for Inductor Cores

The condition of high resistivity requirement for fabrication of inductors can be easily sated using ferrite materials. Ferrites are the compounds of transition metals with oxygen that are ferromagnetic but nonconductive [88]. Soft ferrites having low coercivity are used for inductor cores. Ferrite materials possess high ferromagnetic resonance frequency, and low imaginary permeability and can increase the Q factor in inductors in the high-frequency range. Ferrite-based polymer composites are being researched since the traditional ceramic-based substrates require a very high temperature for processing. At the nanoscale dimensions, the transport, mechanical and electrical properties are dependent on the nanograin boundaries rather than the material and exhibit themselves differently as compared to the bulk material. The major use of ferrites is in radio frequency circuits for receiving and transmission [88].

Ni–Zn–Cu and Co_2_Z (spinal and magnetoplumbite, respectively) nano-powders prepared using the sol-gel method have been used for integration with a single layer and multilayer on-chip conductor. The maximum inductance gain value was found to be 21% for Co_2_Z and 35% for Ni–Zn–Cu in the 10 GHz range in a single layer on-chip inductor integrated with ferrite nanomaterial whereas, for multilayer plus ferrite, the maximum inductance gain was 20 and 25%. The increase in Q factor corresponding to the single layer was 100 and 160% for Co_2_Z and Ni–Zn–Cu, respectively, in the 3 GHz whereas the Q factor increase for the multilayer inductor was 40 and 50% for the same. Integration of a solenoid inductor with nanomaterial led to an increase in inductance by 63 and 168% by Co_2_Z and Ni–Zn–Cu, respectively, at frequency ranges more than 20 GHz. At high frequencies, the Q factor decreases but at lower frequencies, the gain is 71 and 157% with Co_2_Z and Ni–Zn–Cu at 100 MHz [88]. Inductance and Quality factors for various layers of an inductor with ferrite nanomaterial are shown in Figure 6.

Ni–Zn–Cu ferrite nanoparticles-magnetic-core has also been used by Ni et al. in RF ICs. Vertical magnetic cores with multiple-layer stacked-spiral structures were designed to realize compact inductive devices in RF ICs. This design obtained a high L-density of over 700 nH/mm^2^ as well as an 80% chip size reduction with reference to planar inductors. The high performance of the inductor was dependent on both the material and the design [89]. 

Nano-granular Zn_x_Fe_3−x_O_4_ ferrite films have been fabricated on an Ag-coated glass substrate using dimethylamine borane complex -DMAB-Fe (NO_3_)_3_-Zn (NO_3_)_2_ solution. The composition of this film was dependent upon the x value ranging from 0 to 0.99. As the x changes, the microstructure of the film varies from non-uniform nanogranules to fine and uniform nanogranules of 50–60 nm in size. The saturation magnetization was found to be increasing from 75 emu/g to108 emu/g when the x varied from 0 to 0.33. From 0.33, it decreased constantly till reaching 5 emu/g at the x value of 0.99. The coercivity decreased steadily from 116 to 13 Oe [90].

### 4.5. Novel Nanomaterials for Inductor Cores

Recently, carbon nanotubes, nanofibers, nanowires, etc. have been used to create composite polymers for high Q nano inductor applications. Carbon nanotubes (CNT) are preferable for such applications due to their unique interconnecting properties which exhibit lower skin effects as compared to copper interconnects and the resistance of CNT interconnects remains the same even at a high frequency. The inductors made of CNT have several theoretical merits. The magnetic field induced in CNT is 1000 times larger as compared to copper wire and therefore the inductance is quite large. The inductors fabricated from CNTs are smaller as compared to traditional inductors in ICs since they can be bent [91]. Multiwalled carbon nanotube (MWCNT) inductors or single-walled carbon nanotube (SWCNT) bundles have been fabricated and researched by several researchers. The interconnect properties, and electrical and magnetic properties of MWCNTs have been found to be quite high as compared to copper wires. All the research points towards higher inductance, higher Q factor, and lower losses as compared to conventional copper wires [92,93,94,95,96,97]. 

A novel design of MWCNT-based stacked inductors has been described by Bruce C. Kim. The MWCNT-based embedded rf inductors were fabricated on silicon and ceramic substrate which will provide high inductance and quality factors [93]. A high Q nanoinductor based on MWCNT and an introductory layer of nano composite film (Cu/CoFe_2_O_4_) exhibited an inductance of 6.25 nH, with a Q factor of 186 at 2.4 GHz. The maximum inductance and maximum Q factor were found to be 6.6 nH and 440, respectively, and the chip area was reduced by 25% as compared to conventional microscale inductors [97]. Figure 7 shows the inductance and Q factor of the MWCNT inductor with respect to frequency. Wiselin et al. have fabricated a spiral inductor made of carbon nanofibres exhibiting an inductance of 4 nH at 1.2 GHz and a Q factor of 40.5. The self-resonant frequency of the designed inductor is predicted to be in the range of 20 GHz to 30 GHz and since the energy loss is low, the inductor can be used in high-frequency applications [98].

Inductors based on nanowires using nanoporous anodic alumina as substrates have been researched by Spiegel et al. and Hamoir et al. [99,100]. Both have used ferromagnetic Ni nanowires to increase the inductance and quality factor of the inductors. Hamoir et al. have shown a 30% increase in the inductance value and a 23% increase in the quality factor compared to other such ferromagnetic inductors [100]. Spiegel et al. have contrasted the quality factor to that of the inductor built on non-magnetic Si and found the anodic alumina substrate-nanowire-based inductor to be far superior under the FMR frequency. Tunability to magnetic field increases the frequency range and also increases the permeability in the lower frequency ranges [99]. 

Magnetic nanocomposite pastes have also been used for high-frequency properties. Silica-coated cobalt– benzo–cyclo–butane (BCB) and Ni–Zn ferrite epoxy composites have been used to fabricate inductors. These nanocomposites were screen printed onto FR4 substrates and were then patterned using an etch-back process to fabricate inductors. Co–silica–BCB nanocomposite paste was found to have high permittivity and permeability in the range of 5–10 which makes it a candidate for miniaturized antenna fabrication. Inductors made up of Ni–Zn ferrite–epoxy composites exhibited high permeability in the range of 5–15 at low MHz frequencies and 2.5–3 at higher GHz frequencies. The quality factor, Q, was found to be more than 100 for ferrite-epoxy composite samples [101]. 

## 5. Nanomaterials for Interconnect Technology

The role of interconnect in the electrical industry is to connect the active and passive components to the substrate and to facilitate the power supply, and signal transmission among each other, etc. [102]. Traditionally, lead-containing solders most importantly, eutectic tin/lead (Sn/Pb), have been used for electronic packaging owing to their low melting temperatures and good wetting characteristics on a variety of substrates. Chip scale packages, flip chip technologies, etc. are being currently used for interconnections. Apart from the inherent toxicity of lead-based solders, other problems such as incompatibility with miniaturization due to issues of fabrication techniques are making the use of traditional solders difficult. Therefore, there is a need for the development of novel materials which can serve as interconnect in the new upcoming packaging technologies to further increase the miniaturization of the devices. The two major alternatives that have been fueled by the advent of nanotechnology are lead-free solders and polymer-based-electrically-conductive adhesives (ECA) [103,104]. 

### 5.1. Nanoparticle-Based Lead-Free Solder

The major difficulty with lead-free solders is the higher melting temperature required for such solders. This high processing/melting temperature leads to high reflow temperature which in turn builds up the stress and promotes other defects. Nanoparticles provide an alternative to overcome this problem since the particle size in the nanometer range potentially offers a chance for decreased temperature coupled with increased mechanical strength. The addition of nanoparticles to conventional solders may improve their properties. A number of alloys have been considered for the development of lead-free solders mainly involving Sn–Cu, Sn–Ni, and Sn–Ag [105].

Sn–Ag–Cu alloys have been researched extensively for this application. Several researchers are working on reducing the temperature of these alloys either by tinkering with the composition or by developing different fabrication techniques which affect the size dependency of the melting point. Gao et al. have developed a Sn–Ag–Cu lead-free solder alloy having an equivalent melting temperature to that of the Sn–Pb alloy. The nanoparticles in this study were prepared using the consumable direct arc current technique and the nanoparticles developed were in the range of 15–60 nm with an average being 30 nm. This alloy exhibited a melting onset temperature of the nanoparticles of the Sn–Ag–Cu solder as low as 179 °C which is comparable to eutectic Sn–Pb alloys which have melting temperatures in the range of 180–190 °C. Although some nanoparticles did not melt at low temperatures and could potentially be detrimental to this approach, the work nonetheless presents a novel way of reducing the melting temperature [106]. 

Similarly, Zou et al. have studied melting point depression in lead-free Sn–3.0Ag–0.5Cu (wt.%) solder alloy by reducing the size of the particles produced by the direct arc current technique. The melting temperature of both bulk and nanoparticles was examined and the melting point of nanoparticles having an average size of 30 nm showed a depression in the value by 10 °C as compared to bulk alloy [107].

The addition of cobalt (Co) and nickel (Ni) to Sn–Ag–Cu increases the growth of Cu_6_Sn_5_ but reduces the growth of Cu_3_Sn. The addition of both the particles results in the reduction of the interdiffusion coefficient in Cu_3_Sn but had no effect on the melting point of the alloy. These particles did not increase the intermetallic compound thickness and reduced Kirkendall voids, consequently increasing the solder strength and drop test performance. The addition of platinum (Pt) has also been associated with increased drop test performance [108,109]. 

Jiang et al. have studied the size dependency of the melting point in 96.5Sn–3.5Ag alloys. They used a low-temperature chemical reduction method to synthesize variously sized Sn–Ag alloys. Surface stabilizing agents were used to avert the oxidation of the synthesized particles since oxidation does not allow a reduction in temperature. A melting point as low as 194 °C was achieved when the diameter of nanoparticles was around 10 nm. The solder paste composed of synthesized Sn–Ag nanoalloy was used for the wetting test on the Cu substrate and it exhibited a characteristic Cu_6_Sn_5_ intermetallic compound formation. This demonstrated the possibility of Sn–Ag alloy being used as a low-temperature lead-free solder [110] as shown in Figure 8.

Sn–Co–Cu alloy has been developed to overcome the cost of the Sn–Ag–Cu/Sn–Ag alloys. It is being perceived as a cost-effective alternative to the Sn–Ag–Cu alloys. It also suffers from the problem of a high melting point. Sn–0.4Co–0.7Cu (wt.%) lead-free solder alloy was studied by Zou et al. with the aim of decreasing the melting point of the alloy by using nanosized particles. On using the nanoparticle-sized grains from 10 to 50 nm in the alloy, the melting temperature of the alloy was reduced by 5 °C as compared to the bulk alloy. The depression in the melting point was attributed to the increase in the particles’ free energy caused by the reduction in particle size [111]. 

Although a lot of research has been performed on nanoparticles-based lead-free solder materials, their mechanical properties, reliability studies, and processability in packaging technologies still need further elucidation and as such, still, a lot of research is required before this solder replaces the conventional alloys and becomes a reality in the miniaturized digital world.

### 5.2. Electrically Conductive Adhesives

Another class of interconnects that is being researched widely is called electrically conductive adhesives (ECA) [112]. These consist of organic or polymeric matrices which bind metal fillers and form nanocomposites. The metal fillers conduct the electricity and are responsible for the electrical properties, whereas, the matrix is responsible for the mechanical properties. Therefore, ECA provides easy access to controlling the properties of the final product. These adhesives have several advantages over conventional solders such as low processing temperature, lesser fabrication steps, and hence less cost, and fine pitch capacity owing to the presence of conductive fillers. However, they also suffer from a number of drawbacks such as limited electrical and thermal conductivity, a lack of reliability, and poor mechanical strength, comparatively [112,113]. On the basis of the filler loading volume, ECAs are segregated into three parts, isotropically conductive adhesives (ICA): those which provide electrical conductivity in all the three coordinates (x,y,z) due to very high filler content near their percolation threshold, anisotropically conductive adhesives (ACA): those which provide conductivity in just one direction (z-axis) owing to low filler loading and, nonconductive adhesives (NCA): those which provide conductivity in one direction (z-axis) owing to the lack of filler material [114] as shown in Figure 9.

## 6. Isotropically Conductive Adhesives (ICA)

### 6.1. ICAs with Silver Nanowires

ICAs are composed of polymer matrix and conductive filler. Thermosetting and thermoplastic materials are generally used as a polymer. Thermoset epoxies are the most common matrices due to their physical and mechanical properties and thermoplastics are added where strengthening is required for the matrix. The conductive fillers include metals like silver, gold, nickel, etc., and carbon-based materials in different morphologies [116,117]. 

Wu et al. have studied ICA using Ag nanowires as the conductive filler and its properties were compared with conventional ICAs using 1 µm Ag particles and 100 nm Ag particles as the filler. At low filler volumes of Ag nanowires, ICA exhibited lower bulk resistivity and similar shear strength as compared to ICA filled with micrometer and nanometer-sized particles but the shear strength of ICA with particles decreased while trying to match the conductivity of ICA with nanowires. The increase in the conductivity of nanowires was attributed to lower contact resistance, stable network, and tunneling effects [118]. 

### 6.2. ICAs with Silver Nanoparticles

Jiang et al. have prepared ICAs filled with silver nanoparticles. To increase the conductivity of the ICA, Ag nanoparticles were treated with various surfactants. The resistivity of the ICA was reduced to 2.4 × 10^−4^ Ω·cm. The low resistivity was attributed to the sintering of the nanoparticles. In another approach, nanoparticles were surface functionalized using diacids and were then mixed with silver flakes before use as a conductive filler. The use of diacids for surface modification reduced the resistivity to 5 × 10^−6^ Ω·cm. The decreased resistivity was a consequence of the sintering of silver nanoparticles which was confirmed using morphological analysis [119,120]. 

### 6.3. ICAs with Carbon Nanotubes (CNTs)

Multiwalled-carbon-nanotubes (MWCNT) have been used with epoxy resin to formulate ICA. The epoxy-MWCNT composite has a significantly lower percolation threshold as compared to a metal-filled epoxy composite, although, the conductivity of epoxy-MWCNT was of lower magnitude as compared to a metal-filled epoxy composite. The electrical conductivity of the composite is governed by the inherent conductivity and contact resistance of MWCNTs. The shear strength and strength-to-weight ratio of the composite were found to be greater than the metal-filled ICA [121].

Silver-coated carbon nanotubes (SCCNT) have been used as a filler to prepare isotropic conductive adhesives (ICA). The properties of this composite and ICA filled with multiwalled-carbon-nanotubes were compared with 1 µm Ag-filled ICAs. The resistivity value for ICA filled with MWCNTs was found to be 2.4 × 10^−4^ Ω·cm at 31% loading and at filler content of 28%, SCCNT exhibited the lowest conductive resistivity of 2.2 × 10^−4^ Ω·cm. ICAs filled with both MWCNT and SCCNT exhibited a high shear strength value of 19.9 MPa on the Al substrate and 18.2 MPa on the Cu substrate [122]. The relationship between bulk resistivity and shear strength of the ICA with the volume content of CNT is shown in Figure 10.

## 7. Anisotropic Conductive Adhesives/Anisotropic Conductive Films (ACA/ACF)

ACA/ACF is generally used to achieve an ultra-high pitch interconnection. These have been used widely for packaging in liquid crystal displays (LCDs). ACA filled with nano-Ag particles has been studied to understand the effect of nano-sized Ag particles on its capacity. ACA filled with nano-Ag, exhibits reduced joint resistance and increased current carrying capacity. This feature was attributed to the sintering of Ag particles well below their melting temperature and consequently increased the interfacial contact area between the Ag particles and bond pads. Treatment of the nano-Ag particles with self-assembled monolayers of different types helped in further increasing the conductivity by increasing the interfacial contact area to a great extent [123]. 

Li et al. have used a similar treatment of Ag-nano filler by two SAMs, namely dicarboxylic acid, and dithiol. The treatment increased the conductivity of ACAs significantly and the resistance value was decreased from 10^−3^ Ω to 10^−5^ Ω with SAMs-coated silver fillers. This increase in electrical properties was attributed to the bonding between SAM and Ag-nano fillers, consequently increasing the contact interfacial area [124].

### Nonconductive Adhesives (NCA)

Electrically interactive interconnects can be formed by using organic adhesives without any filler. In this method, two contacts are connected using NCA under a moderate value of temperature and pressure. The contact formation depends upon the roughness of the surfaces in contact. Few contact spots are created which allow the current to flow. During the sealing process, pressure is applied, and contacts increase in accordance with the elasticity or flexibility of components. The application of pressure leads to an increase in the interfacial contact area and hence leads to better conductivity. NCA has shown better electrical conductivity than ACA but also suffers from a contact-resistant problem since no metallurgical joints are formed. The nano-sized particles are able to increase the interfacial contact area and reduce the pressure required for fabrication, further increasing the conductivity [115].

## 8. Conclusions

The miniaturization of digital devices using nano-packaging techniques has been hampered by the challenges of the fabrication of nano-sized passive components and interconnect technology. Recent advances in nanotechnology have made it possible to reduce the size of passives to nano-scale by utilizing unique methodologies and materials, thereby increasing the odds of the development of embedded passives, consequently leading to compact and high-quality devices. The denigration of conventional lead-based soldering technology in aspects of health and electrical properties has led to the development of alternate interconnect technology. Nanotechnology has proven its worth in this sector also via the development of nanomaterials-based interconnects having similar or superior properties compared to traditional techniques. In summary, nanotechnology has started to emerge over and above the limitation of books and labs and thus is influencing our lives.

## Figures and Tables

**Figure 1 nanomaterials-12-03284-f001:**
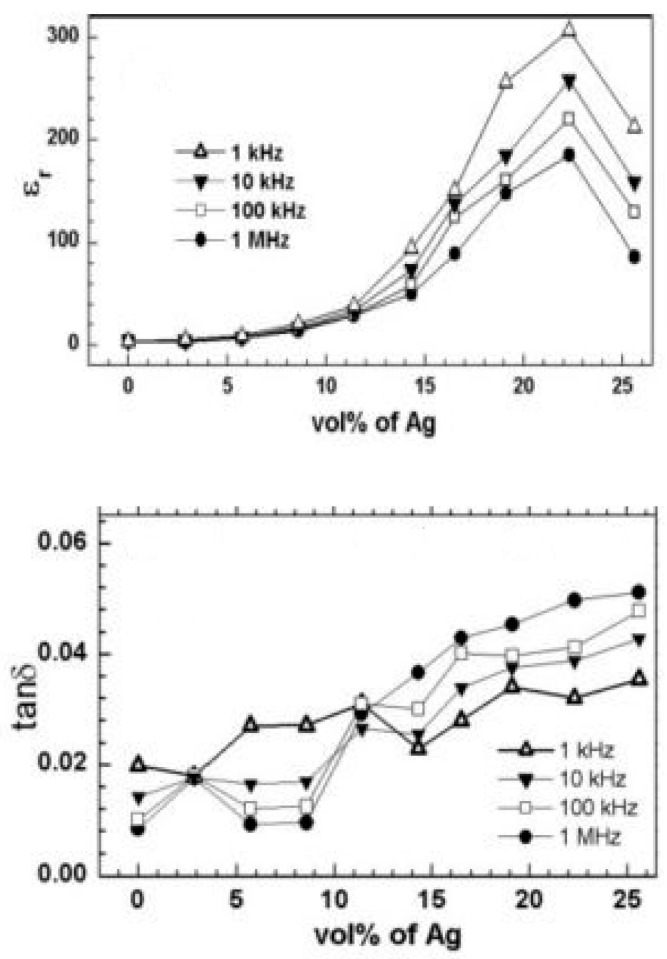
The dependence of dielectric value and dielectric loss on silver volume and frequency. Reprinted with permission from [29]; Copyright 2005, John Wiley and Sons.

**Figure 2 nanomaterials-12-03284-f002:**
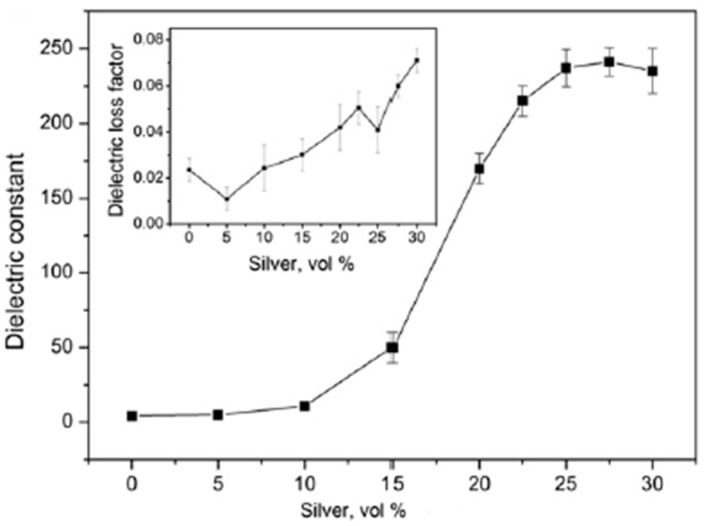
The dependence of dielectric constant and dielectric loss factor with volume fraction of silver. Reprinted with permission from [35]; Copyright 2012, Elsevier.

**Figure 3 nanomaterials-12-03284-f003:**
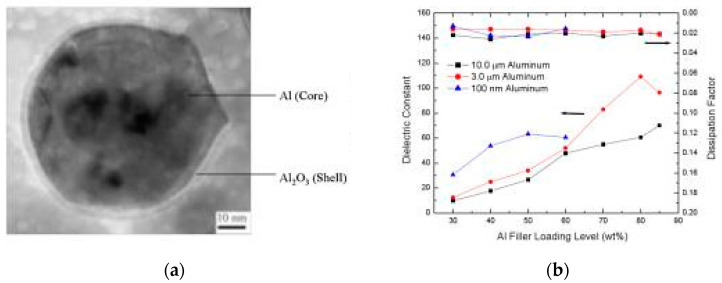
(**a**) TEM image of Al nanoparticle with Al oxide coating (**b**) dependence of dielectric constant on filler loading volume. Reprinted with permission from [37]; Copyright 2005, AIP Publishing.

**Figure 4 nanomaterials-12-03284-f004:**
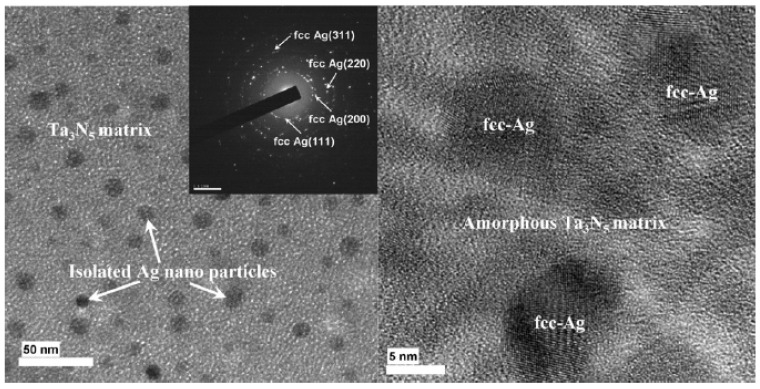
Ag nanocomposite thin film deposited at 55% N_2_ partial pressure. Reprinted with permission from [56]; Copyright 2008, Elsevier.

**Figure 5 nanomaterials-12-03284-f005:**
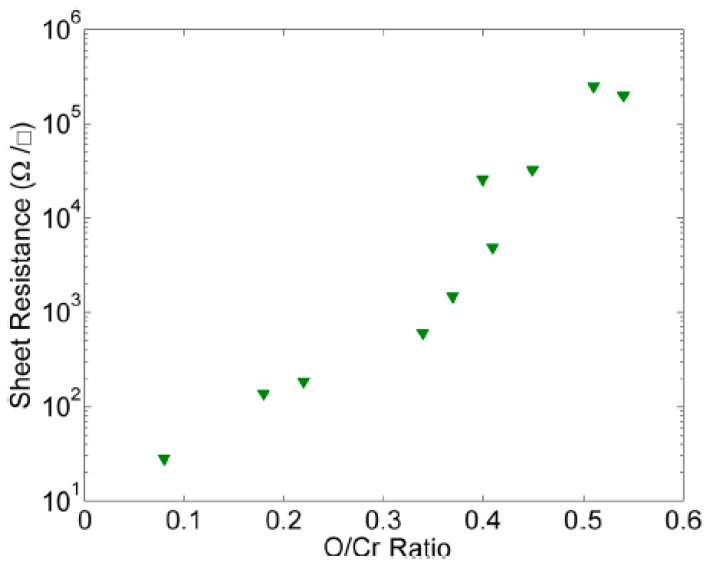
Dependence of sheet resistance of chromium oxide films with the oxygen-to-chromium mass ratio at room temperature. Reprinted with permission from [59]; Copyright 2014, AIP Publishing.

**Figure 6 nanomaterials-12-03284-f006:**
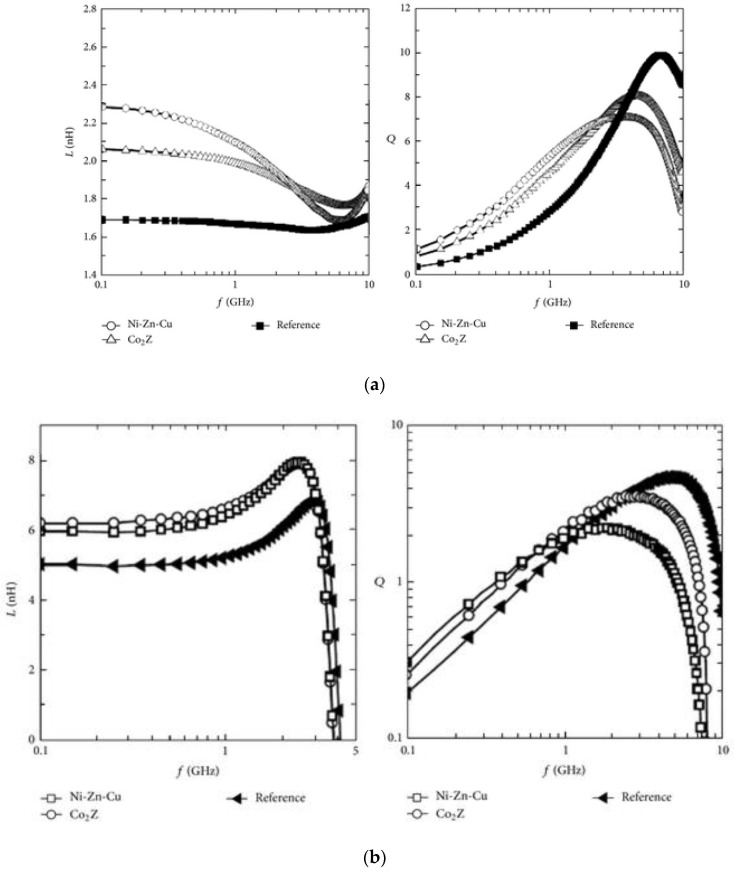
Inductance and Quality factor for (**a**) single layer, (**b**) multilayer, and (**c**) solenoid inductor with ferrite nanomaterial. Reprinted with permission from [88].

**Figure 7 nanomaterials-12-03284-f007:**
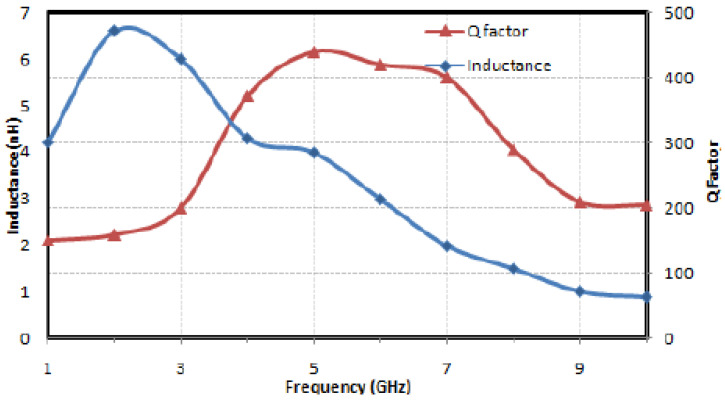
Inductance and Q factor of MWCNT inductor with respect to frequency. Reprinted with permission from [97].

**Figure 8 nanomaterials-12-03284-f008:**
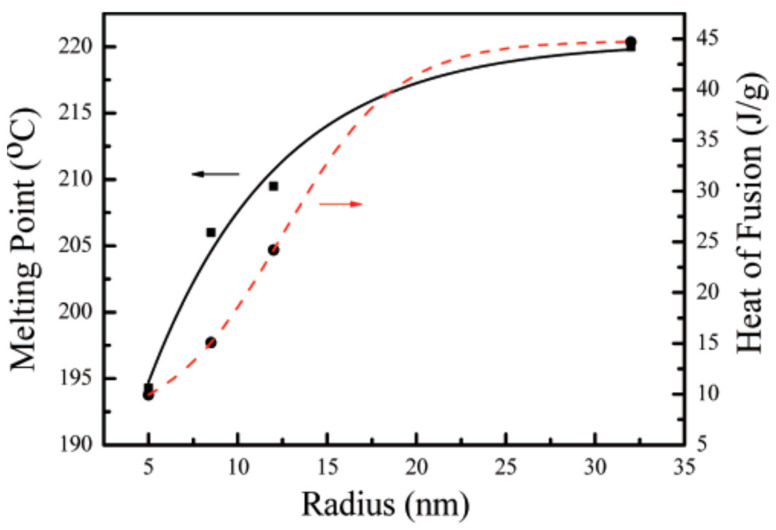
Melting point: heat of fusion dependence on the radius of the synthesized Sn–Ag nanoparticles. Reprinted with permission from [110]; Copyright 2007, American Chemical Society.

**Figure 9 nanomaterials-12-03284-f009:**
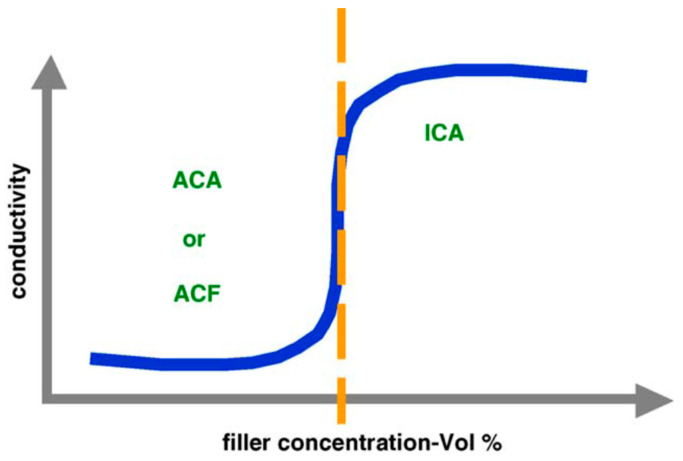
Curve representing the relationship conductivity and filler concentration with respect to percolation threshold. Reprinted with permission from [115]; Copyright 2006, Elsevier.

**Figure 10 nanomaterials-12-03284-f010:**
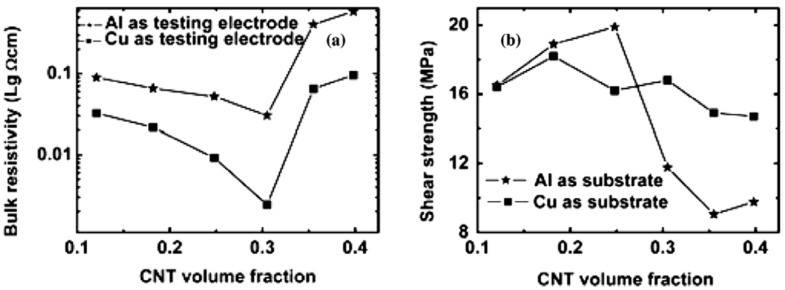
(**a**,**b**) The relationship between bulk resistivity, shear strength of ICA with the volume content of CNT. Reprinted with permission from [122]; Copyright 2007, Elsevier.

## Data Availability

The raw data supporting the conclusions of this article will be made available by the authors, without undue reservation.

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
