# Peer review of "Development and Applications of Embedded Passives and Interconnects Employing Nanomaterials"

_nanomaterials, 2022, doi:10.3390/nano12193284_

Round 1

Reviewer 1 Report

Thanks for interesting review

Author Response

The revisions have been made please find the attached file. 

Reviewer 2 Report

Dear authors,

thank you for this overview of nanomaterials.

First some minor corrections;

The very first phrase in the 3.3 chapter needs to be reformulated "The second category amongst the passive components that are being researched 
consistently for reduction in the size of the printed circuit board/printed wiring board (PCB/PWB) is the resistor"

Chapter 3.7, first paragraph, 2nd line. There is a space too much.

Chapter 3.7, third paragraph replace "enourmously" by "thoroughly", or by

  1. In a thorough manner; fully; entirely; completely.
  2. In a thorough or complete manner.
  3. in an exhaustive manner

Chapter 5.1 second paragraph replace "Sn-Ag- Cu" by "Sn-Ag-Cu" (there is a space before Cu to be deleted)

Now on understanding:

1) these embedded passives should be embedded where? not in the semiconductor, but in the PCB? It is not explained how to use these nanomaterials in PCB-fabrication. From the introduction I had a feeling you would talk about passives implemented in ICs but then it seems more to be on PCBs.

2) if used on PCB: how can sputtering or other deposit methods be used with such large sheets?

3) the different nanomaterials wether for R/L or C-implementation are not compared to what is used today, so it is hard to see if they are better/advantageuos or not

4) in the beginning of chapter 3.3 you say that "Embedded resistors will also be able to increase the reliability and electrical performance of the circuit". Id' like to know howcome?

5) so these nanomaterials should replace SMDs, what would be the real estate gain?

6) chapter 4.2, second paragraph. You say "Co-Zr-O alloys exhibit preferable properties on water cooled substrates which makes it suitable for embedded inductor applications." Please explain, why?

7) chapter 5.2 you say that the NCA provides conductivity in one direction, I thought that was ACA, please explain better howcome NCAs are conductive

8) your figure 10 includes ACF which has not been explained yet in the text. Why is this figure relevant?

9) chapter 7.1 - again the NCAs that have electrical conductivity...is it used for pressure induced contact?

10) it seems that the illustrations come from other publications, cite them in the figure text, and permission to republish should be asked at the different editors office. The quality of some (most) of the illustrations is not very good, could it be improved?

Thank you and have a nice day

Author Response

The revision has been made please find the attached file. 

Reviewer 3 Report

Review comments for

Manuscript ID: nanomaterials-1698442-peer-review-v1      

Title: Development and Applications of Embedded Passives and Interconnects Employing Nanomaterials

Shanggui Deng, Sharad Bhatnagar, Shan He *, Nabeel Ahmad *, Abdul Rahaman, Jingrong Gao, Jagriti Narang, Ibrahim Khalifa, Anindya Nag

Submitted to: Nanomaterials

Comments:

The submitted manuscript claims to provide a comprehensive review of embedded passive and interconnects using nanomaterials.

It has the following sections:

1)      Embedded Capacitors

2)      Ferroelectric Ceramic- Polymer Composites

3)      Alloy based nanomaterials for inductor cores

4)      Nanomaterials for Interconnect Technology

5)      Isotropically conductive adhesives (ICA)

6)      Anisotropic conductive adhesives/ anisotropic conductive films (ACA/ACF)

However, the submitted manuscript only picked up a few examples in each section and provided details. The current version of the manuscript is confusing and does not provide a complete picture to readers to determine the leading performing materials compare to their competitors in each section individually. For example, in the “section embedded capacitors”, the obvious question to a reader would be: who is the best material choice in this section currently and why? Furthermore, who would be the leading material in this section in the near future in terms of the performance? The current version of the manuscript does not provide any answer to this basic question.

Therefore the following items should be added to each section individually.

1)      Technology roadmaps

2)      Historical milestones

3)      Performance comparison table/graphs

4)      Section leaders and their advantages, disadvantages

5)      Section bottlenecks

6)      Future perspectives

These are completely absent in the current version and can not justify the importance of the manuscript. Therefore, the current form of the manuscript is premature in its current format and needs a major revision.

Author Response

The revisions have been done as per the comment please find the attached file. 

Round 2

Reviewer 3 Report

The authors have modified the manuscript accordingly. The modifications improved the manuscript quality and readability. The review article is appropriately represented and easily understandable. I recommend the current version of the manuscript for publication.